# M³RL: Mind-aware Multi-agent Management Reinforcement Learning

**Tianmin Shu** *
University of California, Los Angeles
`tianmin.shu@ucla.edu`

**Yuandong Tian**
Facebook AI Research
`yuandong@fb.com`

## Abstract

Most of the prior work on multi-agent reinforcement learning (MARL) achieves optimal collaboration by directly learning a policy for each agent to maximize a common reward. In this paper, we aim to address this from a different angle. In particular, we consider scenarios where there are self-interested agents (i.e., worker agents) which have their own minds (preferences, intentions, skills, etc.) and can not be dictated to perform tasks they do not want to do. For achieving optimal coordination among these agents, we train a super agent (i.e., the manager) to manage them by first inferring their minds based on both current and past observations and then initiating contracts to assign suitable tasks to workers and promise to reward them with corresponding bonuses so that they will agree to work together. The objective of the manager is to maximize the overall productivity as well as minimize payments made to the workers for ad-hoc worker teaming. To train the manager, we propose Mind-aware Multi-agent Management Reinforcement Learning (M³RL), which consists of agent modeling and policy learning. We have evaluated our approach in two environments, Resource Collection and Crafting, to simulate multi-agent management problems with various task settings and multiple designs for the worker agents. The experimental results have validated the effectiveness of our approach in modeling worker agents' minds online, and in achieving optimal ad-hoc teaming with good generalization and fast adaptation.[1]

## 1 Introduction

As the main assumption and building block in economics, self-interested agents play a central roles in our daily life. Selfish agents, with their private beliefs, preferences, intentions, and skills, could collaborate (*ad-hoc teaming*) effectively to make great achievement with proper incentives and contracts, an amazing phenomenon that happens every day in every corner of the world.

However, most existing multi-agent reinforcement learning (MARL) methods focus on collaboration when agents selflessly share a common goal, expose its complete states and are willing to be trained towards the goal. While this is plausible in certain games, few papers address the more practical situations, in which agents are self-interested and inclined to show off, and only get motivated to work with proper incentives.

In this paper, we try to model such behaviors. We have multiple workers and a manager, together to work on a set of tasks. The manager gets an external reward upon the completion of some tasks, or one specific task. Each worker has a skill set and preference over the tasks. Note that their skills and preferences may not align with each other (Fig. 1(a)), and are not known to the manager (Fig. 1(b)). Furthermore, manager may not get any external reward until a specific task is complete, which depends on other tasks.

By default, the self-interested workers simply choose the most preferred tasks, which is often unproductive from the perspective of the entire project. Therefore, the manager gives additional incentives in the form of *contracts*. Each contract assigns a goal and a bonus for achieving the goal to a worker.

---

*Work done while interning at Facebook AI Research.
[1]Code is available at `https://github.com/facebookresearch/M3RL`.

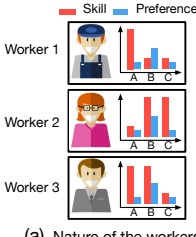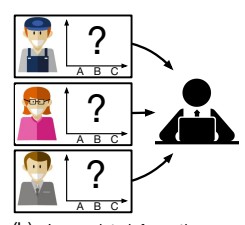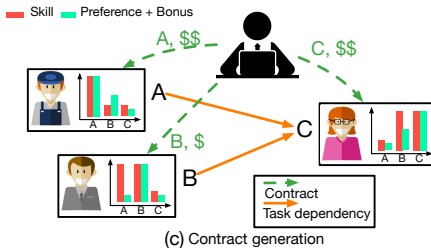

(a) Nature of the workers    (b) Incomplete information    (c) Contract generation

Figure 1: Illustration of our problem setup. Workers have different skills (abilities for completing tasks) and preferences (which tasks they like) indicated by the bar charts. They are self-interested and perform the tasks they prefer the most. To achieve optimal collaboration, a manager has to first infer workers' minds, and assigns right bonuses to workers for finishing specified tasks in the form of contracts. Consequently, workers will adjust their intentions and work together accordingly. E.g., workers in the figure initially all want to do task B. To finish all tasks, the manager has to pay more bonus to worker 1 and 2 so that they will perform A and C respectively.

With the external incentives, workers may choose different goals than their preferences. Upon completion of assigned goals, the manager receives the rewards associated with those goals and makes the promised payments to the workers. To generate optimal contracts, the manager must infer the workers' minds and learn a good policy of goal and reward assignment.

Conventional approaches of mechanism design tackle similar problems by imposing strong assumptions (e.g., skill/preference distributions, task dependencies, etc) to find an analytic solution. In contrast, we aim to train a *manager* using reinforcement learning to **i)** assess minds of workers (skills, preferences, intentions, etc.) on the fly, **ii)** to optimally assign contracts to maximize a collaborative reward, and **iii)** is adapted to diverse and even evolving workers and environments.

For this, we propose a novel framework – Mind-aware Multi-agent Management Reinforcement Learning (M$^3$RL), which entails both agent modeling for estimating workers' minds and policy learning for contract generation. For agent modeling, we infer workers' identities by their performance history, and track their internal states with a mind tracker trained by imitation learning (IL). For contract generation, we apply deep reinforcement learning (RL) to learn goal and bonus assignment policies. To improve the learning efficiency and adaptation, we also propose high-level successor representation (SR) learning (Kulkarni et al., 2016) and agent-wise $\epsilon$-greedy exploration.

As a proof of concept, we evaluate our approach in two environments: Resource Collection and Crafting in 2D Minecraft, to simulate multi-agent management problems. The setup and underlying assumptions are designed to mimic real world problems, where workers are not compelled to reveal their true preferences and skills, and there may be dependency between tasks resulting in delayed and sparse reward signals. Workers may also be deceitful (e.g., accepting a contract even when the assigned goal is unreachable). Our experiments demonstrate that the manager trained by our approach can **i)** estimate the mind of each worker from the recent behaviors, **ii)** motivate the workers to finish less preferable or intermediate tasks by assigning the right bonuses, **iii)** is adaptive to changing teams, e.g., change of members and/or change of workers' skills and preferences, **iv)** and has good generalization in different team sizes and novel environments.

We have conducted substantial ablation studies by removing the key components, including IL, SR, agent-wise $\epsilon$-greedy exploration, and performance history. Our approach shows a consistent performance in standard settings as well as in more challenging ones where workers' policies are stochastic and sub-optimal, or there are multiple levels of bonuses required to motivate workers.

## 2    RELATED WORK

**Multi-agent reinforcement learning**. For collaboration problems, common multi-agent reinforcement learning (Littman, 1994; Busoniu et al., 2008) usually trains agents (Oliehoek et al., 2008; Foerster et al., 2016; Peng et al., 2017; Omidshafiei et al., 2017; Lowe et al., 2017) so that they will jointly maximize a shared reward. There also have been work on contributing different credits to agents by factorized value functions (Koller & Parr, 1999; Guestrin et al., 2001; Sunehag et al., 2018;

Rashid et al., 2018), but the spontaneous collaboration assumption is still required. In contrast, we instead train a manager to manage multiple self-interested workers for an optimal collaboration.

**Principal-agent problems**. Our problem setup is closely related to principal-agent problems (Laffont & Martimort, 2002) (or moral hazard problems (Hlmstrom, 1979)) in economics. Our manager and workers can be considered as the principal and agents respectively, where agents and principal have different objectives, and the principal needs to provide the right incentives to ensure that the agents make the best choices for what the principal delegates. These problems face similar technical challenges as our problem setup, e.g., information asymmetry between principals and agents, how to setup incentive cost, how to infer agents types, how to monitor their behaviors, etc. Traditional approaches in economics (Myerson, 1982; Hlmstrom & Milgrom, 1991; Sannikov, 2008) build mathematical models to address these issues separately in stateless games, often with the assumption that the utility functions and the behavior patterns of the agents are known, leading to complicated models with many tunable parameters. In comparison, our paper provides a practical end-to-end computational framework to address this problem in a data-driven way without any assumption about the agents' utilities and their decision making processes. Moreover, this framework is adaptive to changes of agents preferences and capabilities, which very few papers in economics have addressed. We also evaluate our approach in more complex game settings than the ones in the current economics literature.

**Mechanism design**. Similar to our problem setting, mechanism design also tackles problems where agents have different and private preferences (Myerson, 1981; Conitzer & Sandholm, 2002). Its core idea is to set up rules so that the agents will truthfully reveal their preferences for their own interests, and ultimately an optimal collective outcome can be achieved. Our work differs from mechanism design in several ways. First, in addition to preferences, we also acknowledge the fact that agents may have different skills. Second, mechanism design does not consider sequential decision problems, whereas we have to dynamically change the contracts over time.

**Optimal reward design**. The contract generation in our work can be seen as reward design. Some prior work has proposed optimal reward design approaches (Zhang et al., 2009; Zhang & Parkes, 2008; Sorg et al., 2010; Ratner et al., 2018), where a teacher designs the best reward so that the student will learn faster or alter its policy towards the target policy. In contrast, we try to use deep RL to train optimal reward design policies to manage multi-agents in more complex tasks.

**Meta-learning**. Our work also resembles meta-learning (Wang et al., 2016; Finn et al., 2017), which typically aims at learning a meta strategy for multiple tasks (Maclaurin et al., 2015; Duan et al., 2017; Hariharan & Girshick, 2017; Wichrowska et al., 2017; Yu et al., 2018; Baker et al., 2017) with good sample efficiency, or for a fast adaptation (Al-Shedivat et al., 2018). The meta-learning in this paper is for addressing the problem of ad-hoc teaming (Bowling & McCracken, 2005; Stone et al., 2010) by training from a limited set of worker population.

**Theory of Mind**. Our agent modeling is inspired by the prior work on computational theory of mind, where both Bayesian inference (Baker et al., 2009) and end-to-end training (Rabinowitz et al., 2018) have been applied to understand a single agent's decision making by inferring their minds. In this work, we extend this to optimal multi-agent management by understanding agents' minds.

## 3 PROBLEM SETUP

In an environment, there is a set of goals $\mathcal{G}$ corresponding to several tasks, $N$ self-interested workers with different minds, and a manager which can observe workers' behaviors but is agnostic of their true minds. Different from the common Markov game setting for MARL in prior work (Littman, 1994), we use an independent Markov Decision Process (MDP), i.e., $\langle \mathcal{S}_i, \mathcal{A}_i, R_i, \mathcal{T}_i \rangle$, $\forall i \in N$, to model each worker, where $\mathcal{S}_i$ and $\mathcal{A}_i$ are the state space and action space, $R_i : \mathcal{S}_i \times \mathcal{G}_i \to \mathbb{R}$ is the reward function, and $\mathcal{T}_i : \mathcal{S}_i \times \mathcal{A}_i \to \mathcal{S}_i$ is the state transition probabilities. For achieving goals, a worker has its own policy $\pi_i : \mathcal{S}_i \times \mathcal{G}_i \to \mathcal{A}_i$. We define the key concepts in this work as follows.

**Contract**. A contract is a combination of goal and bonus assignment initiated by the manager to a specific worker. For simplicity, we consider discrete bonuses sampled from a finite set $\mathcal{B}$. Thus, for worker $i$ at time $t$, it will receive a contract defined as $(g_i^t, b_i^t)$, where $g_i^t \in \mathcal{G}$ is the goal and $b_i^t \in \mathcal{B}$ is the corresponding bonus for achieving the goal. Note that the contract will change over time.

**Worker's mind**. We model a worker's mind by its preferences, intentions, and skills. We do not study worker agents' beliefs in this paper, which we leave as future work.

**Preference**. A worker's preference is formally defined as its bounded internal utilities of achieving different goals, $\boldsymbol{u}_i = (u_{ig} : g \in \mathcal{G})$, where $0 \leq u_{ig} \leq u_{\max}$. Combined with received contract, the worker agent's reward function can be defined as

$$r_{ig}^t = R_i(s_i^t, g) = (u_{ig} + \mathbb{1}(g = g_i^t)b_i^t)\mathbb{1}(s_i^t = s_g), \quad g \in \mathcal{G}. \tag{1}$$

where $s_g$ is the goal state.

**Intention**. The intention of a worker is the goal it is pursuing at any time, i.e., $\mathcal{I}_i^t \in \mathcal{G}$, which is not fully revealed to the manager. Based on the reward defined in Eq. (1), there are multiple ways to choose the goal. For a rational worker who is clear about its skills, it will choose the goal by maximizing expected return. I.e., $\mathcal{I}_i^t = \arg\max_g \mathbb{E}[\sum_{t=0}^{\infty} \gamma_i^t r_{ig}^t]$, where $0 < \gamma_i \leq 1$ is its discount factor. However, this requires a worker to have a good estimate of its skills and to be honest, which is not always true. E.g., a worker may want to pursue some valuable goal that it can not reach. So an alternative way is to maximize the utility instead: $\mathcal{I}_i^t = \arg\max_g u_{ig} + \mathbb{1}(g = g_i^t)b_i^t$. This will make a worker's behavior more deceptive as it may agree to pursue a goal but will rarely produce a fruitful result. In this work, we focus on the second way to achieve a more realistic simulation. After determine which goal to pursue, a worker will decide whether to sign the assigned contact. We denote this by $d_i^t \in \{0, 1\}$, where $d_i^t = 1$ means that worker $i$ signs the contract given at time $t$.

**Skill**. The skill of a worker is jointly determined by its state transition probabilities $\mathcal{T}_i$ and its policy conditioned on its intention, i.e., $\pi_i(\cdot|s_i^t, \mathcal{I}_i^t)$.

**Manager's objective**. The manager in our setting has its own utility $\boldsymbol{v} = (v_g : g \in \mathcal{G})$, where $v_g \geq 0$ is the utility of achieving goal $g$. To maximize its gain, the manager needs to assign contracts to workers optimally. For the sake of realism, we do not assume that the manager knows for sure if a worker agent is really committed to the assignment. The only way to confirm this is to check whether the goal achieved by the worker is consistent with its last assigned goal. If so, then the manager will gain certain reward based on its utility of that goal and pay the promised bonus to the worker. Thus, we may define the manager's reward function as:

$$r^t = R^M(S^{t+1}) = \sum_{g \in \mathcal{G}} \sum_{i=1}^{N} \mathbb{1}(s_i^{t+1} = s_g)\mathbb{1}(g = g_i^t)(v_g - b_i^t), \tag{2}$$

where $S^{t+1} = \{s_i^{t+1} : i = 1, \cdots, N\}$ is the collective states of all present worker agents at time $t + 1$. The objective of the manager is to find optimal contract generation to maximize its expected return $\mathbb{E}[\sum_{t=0}^{\infty} \gamma^t r^t]$, where $0 < \gamma \leq 1$ is the discount factor for the manager. Note that the manager may get the reward of a goal for multiple times if workers reach the goal respectively.

**Population of worker agents**. The trained manager should be able to manage an arbitrary composition of workers rather than only specific teams of workers. For this, we maintain a population of worker agents during training, and sample several ones from that population in each episode as the present workers in each episode. The identities of these workers are tracked across episodes. In testing, we will sample workers from a new population that has not been seen in training.

## 4 APPROACH

Our approach has three main components as shown in Figure 2: i) performance history module for identification, ii) mind tracker module for agent modeling, and iii) manager module for learning goal and bonus assignment policies. We introduce the details of these three components as follows.

### 4.1 PERFORMANCE HISTORY MODULE AND MIND TRACKER MODULE

To model a worker's mind, we first need to infer its identity so that the manager can distinguish it from other agents. Previous work (Rabinowitz et al., 2018) typically identifies agents via their trajectories in recent episodes. This only works when diverse past trajectories of agents are available beforehand. However, this is impractical in our problem as the past trajectories of a worker depends on the manager's policy, and thus are highly correlated and can hardly cover all aspects of that agent.

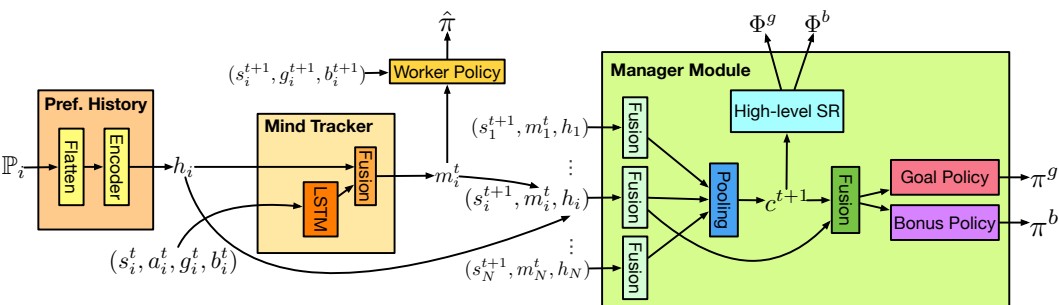

Figure 2: Overview of our network architecture.

In this work, we propose performance history for agent identification, which is inspired by the upper confidence bound (UCB) algorithm (Auer et al., 2002) for multi-bandit arm (MAB) problems. Formally, the performance history of worker $i$ is a set of matrices $\mathbb{P}_i = \{\boldsymbol{P}_i^t = (\rho_{igb}^t) : t = 1, \cdots, T\}$, where $0 \leq \rho_{igb}^t \leq 1$ is an empirical estimation of the probability of worker $i$ finishing goal $g$ within $t$ steps after signing the contract if promised with a bonus of $b$. We discuss how to update this estimate in Algorithm 1. These matrices are then flatten into a vector and we encode it to a history representation, $h_i$, for worker $i$.

With identification, the manager uses an independent mind tracker module with shared weights to update its belief of a worker's current mental state online by encoding both current and past information: $M(\Gamma_i^t, h_i)$, where $\Gamma_i^t = \{(s_i^\tau, a_i^\tau, g_i^\tau, b_i^\tau) : \tau = 1, \cdots, t\}$ is a trajectory of the worker's behavior and the contracts it has received upon current time $t$ in the current episode.

## 4.2 MANAGER MODULE

For contract generation, the manager has to consider all present workers as a context. Thus, we encode each worker's information and pool them over to obtain a context representation, i.e., $c^{t+1} = C(\{(s_i^{t+1}, m_i^t, h_i) : i = 1, \ldots, N\})$. With both individual information and the context, we define goal policy, $\pi^g(\cdot|s_i^{t+1}, m_i^t, h_i, c^{t+1})$, and bonus policy, $\pi^b(\cdot|s_i^{t+1}, m_i^t, h_i, c^{t+1})$, for each worker.

In addition to learning policies for individual workers, we also want the manager to estimate the overall productivity of a team. A common choice in previous literature (e.g., Lowe et al. (2017)) is to directly learn a centralized value function based on the context. However, this is not informative in our case, as the final return depends on achieving multiple goals and paying different bonuses. It is necessary to disentangle goal achievements, bonus payments, and the final net gain.

To this end, we adopt the idea of successor representation (SR) (Kulkarni et al., 2016; Zhu et al., 2017; Barreto et al., 2017; Ma et al., 2018), but use it to estimate the expectation of accumulated goal achievement and bonus payment in the future instead of expected state visitation. By defining two vectors $\phi^g(c^t)$ and $\phi^b(c^t)$ indicating goal achievement and bonus payment at time $t$ respectively, we may define our high-level SR, $\Phi^g$ and $\Phi^b$, as $\Phi^g(c^t) = \mathbb{E}[\sum_{\tau=0}^{\infty} \gamma^\tau \phi^g(c^{t+\tau})]$ and $\Phi^b(c^t) = \mathbb{E}[\sum_{\tau=0}^{\infty} \gamma^\tau \phi^b(c^{t+\tau})]$. We discuss the details in Appendix A.1.

## 4.3 LEARNING

For a joint training of these three modules, we use advantage actor-critic (A2C) (Mnih et al., 2016) to conduct on-policy updates, and learn SR similar to Kulkarni et al. (2016). In addition, we also use imitation learning (IL) to improve the mind tracker. In particular, we predict a worker's policy based on its mental state representation, i.e., $\hat{\pi}(\cdot|s_i^t, g_i^t, b_i^t, m_i^{t-1})$, which is learned by an additional cross-entropy loss for action prediction. Section A.2 summarizes the details. As our experimental results in Section 5 and Appendix C show, in difficult settings such as random preferences and multiple bonus levels, the policies based on the mental state representation trained with IL have a much better performance than the ones without it.

As the manager is agnostic of workers' minds, it is important to equip the manager with a good exploration strategy to fully understand each worker's skills and preferences. A common exploration

strategy in RL is $\epsilon$-greedy, where an agent has a chance of $\epsilon$ to take random actions. However, this may cause premature ending of contracts where a worker does not have sufficient amount of time to accomplish anything. Therefore, we adopt an agent-wise $\epsilon$-greedy exploration, where a worker has as a chance of $\epsilon$ to be assigned with a random goal at the beginning of an episode and the manager will never change that goal assignment throughout the whole episode. In this way, it is easier for a manager to understand why or why not a worker is able to reach an assigned goal. The details can be seen from the rollout procedure (Algorithm 1) in Appendix B.

## 5 EXPERIMENTS

### 5.1 GENERAL TASK SETTINGS

We introduce the general task settings as follows. Note that without additional specification, workers are implemented as rule-based agents (detailed in Appendix D.2).

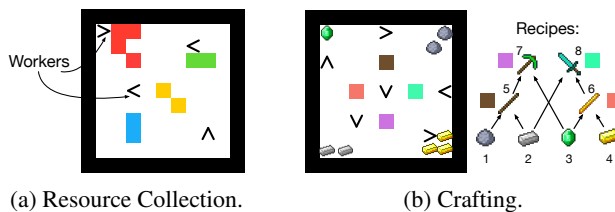

(a) Resource Collection.  (b) Crafting.

Figure 3: (a) Resource Collection environment, where the colored blocks are the resources and the arrows are the workers. (b) Crafting environment (left) and the recipe (right), where the numbers indicate item categories, and the colored block beside an item shows where this item can be crafted.

### 5.1.1 RESOURCE COLLECTION

In Resource Collection, the goals are defined as collecting certain type of resources. There are 4 types of resources on a map (Figure 3a) and the total quantity is 10. A worker can find any resources but only has the skills to dig out certain types of resources. Note that it may not be skilled at collecting its preferred resources. We consider three different settings:

- S1: Each agent can collect up to three types of resources including its preferred type.
- S2: Each agent can only collect one type of resource which may or may not be its preferred one.
- S3: Similar to S2, except that an agent has a different random preference in each episode and thus its preference can not be inferred from history.

A worker can take five actions: "move forward", "turn left", "turn right", "collect", and "stop", and its skill is reflected by the effect of taking the "collect' action. For workers, the internal utility of a resource is 1 if it is preferred; otherwise it is 0. The manager receives a reward of 3 for every resource collected under the contracts, and can choose to pay a worker with a bonus of 1 or 2.

### 5.1.2 CRAFTING

Different from previous work (Andreas et al., 2017) where all items can be directly crafted from raw materials, we consider three-level recipes (Figure 3b): crafting a top-level item requires crafting certain intermediate item first. There are four work stations (colored blocks) for crafting the four types of items respectively. For the manager, each top-level item is worth a reward of 10, but collecting raw materials and crafting intermediate items do not have any reward. Note that certain materials are needed for crafting both top-level items, so the manager must strategically choose which one to craft. In each episode, there are raw materials sufficient for crafting one to two top-level items. All collected materials and crafted items are shared in a common inventory.

We define 8 goals including collecting raw materials and crafting items. Each worker prefers one of the collecting goals (the internal utility is 1), and is only capable of crafting one type of items. We

expands the action space in Section 5.1.1 to include "craft", which will only take effect if it has the ability of crafting the intended item and there are sufficient materials and/or intermediate items. The manager can choose a bonus from 0 to 2 for the contracts, where 0 means no employment.

## 5.2 BASELINES

For comparison, we have evaluated the following baselines:

- Ours w/o SR: Learning a value function directly w/o successor representations.
- Ours w/o IL: Removing action prediction loss.
- Temporal $\epsilon$-greedy: Replacing the agent-wise exploration with conventional $\epsilon$-greedy exploration.
- Agent identification using recent trajectories: Encoding an agent's trajectories in the most recent 20 episodes instead of its performance history, which is adopted from Rabinowitz et al. (2018).
- UCB: Applying UCB (Auer et al., 2002) by defining the management problem as $N$ multi-armed bandit sub-problems, each of which is for a worker agent. In each MAB sub-problem, pulling an arm is equivalent to assigning a specific goal and payment combination to a worker agent (i.e., there are $|\mathcal{G}| \cdot |\mathcal{B}|$ arms for each worker agent).
- GT types known: Revealing the ground-truth skill and preference of each worker and removing the performance history module, which serves as an estimation of the upper bound performance.

## 5.3 LEARNING EFFICIENCY

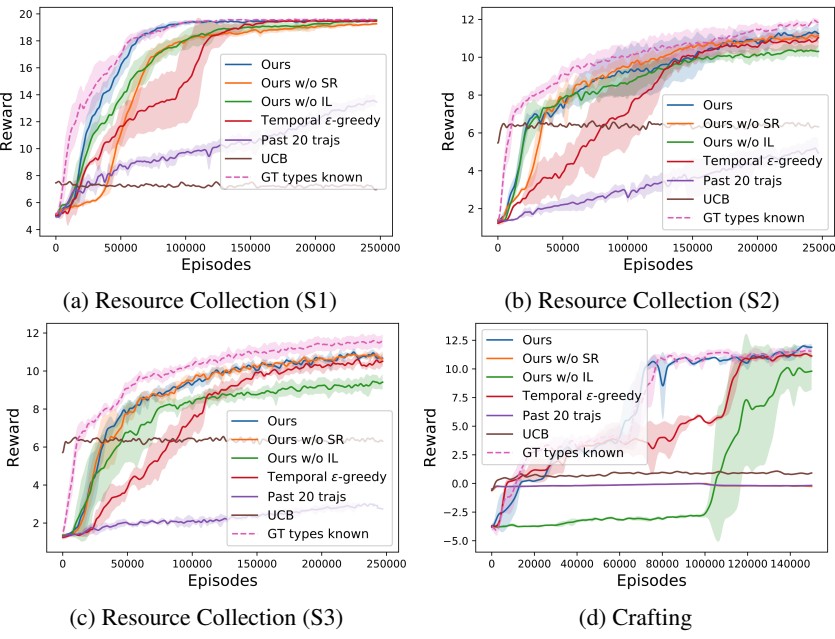

(a) Resource Collection (S1)

(b) Resource Collection (S2)

(c) Resource Collection (S3)

(d) Crafting

Figure 4: Learning curves of all approaches in Resource Collection and Crafting. The rewards here are not rescaled and we show results from 5 runs in all experiments.

During training, we maintain a population of 40 worker agents. In each episode, we sample a few of them (4 workers in Resource Collection and 8 workers in Crafting). All approaches we have evaluated follow the same training protocol. The learning curves shown in Figure 4 demonstrate that ours consistently performs the best in all settings, and its converged rewards are comparable to the one trained using ground-truth agent types as part of the observations. Moreover, in more difficult settings, e.g., S3 of Resource Collection and Crafting, the benefits of IL, SR, agent-wise $\epsilon$-greedy exploration, and the history representations based on the performance history are more significant. In particular, when there are tasks that do not have any reward themselves such as in Crafting, SR and IL appear to offer the most critical contributions. Without them, the network hardly

gets any training signals. In all cases, the agent identification by encoding recent trajectories learns extremely slowly in Resource Collection and fails to learn anything at all in Crafting.

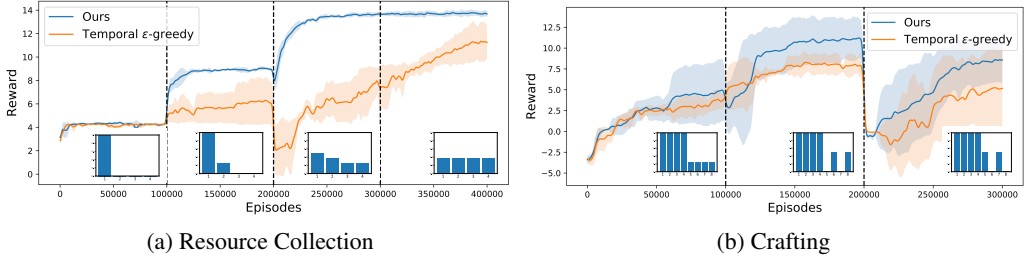

(a) Resource Collection

(b) Crafting

Figure 5: Comparison of the adaption capabilities of different exploration strategies during training. The dashed lines indicate the changing points of the worker agents' skills. The histograms show how the skill distribution in the same population evolve over time.

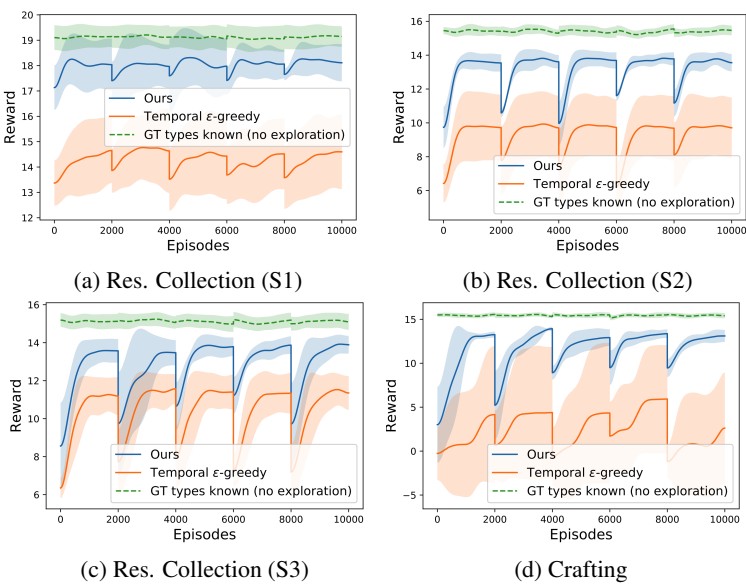

(a) Res. Collection (S1)

(b) Res. Collection (S2)

(c) Res. Collection (S3)

(d) Crafting

Figure 6: Testing performance when old team members are constantly replaced by new ones.

## 5.4 ADAPTATION AND GENERALIZATION

In real world scenarios, the population of worker agents and their skills may evolve over time, which requires the manager to continuously and quickly adapt its policy to the unforeseeable changes through a good exploration. Thus we compare our agent-wise $\epsilon$-greedy exploration with the temporal $\epsilon$-greedy exploration in two cases: i) training with a population where workers' skills change drastically after 100,000 episodes (the manager does not know when and which workers' skill sets have been updated), and ii) testing with a team where 75% of the workers will be replaced with new ones after every 2,000 episodes. Both strategies keep the same constant exploration coefficient, i.e., $\epsilon = 0.1$. To have a better sense of the upper bound in the testing case, we also show the performance of the baseline that knows ground-truth agent information where no exploration is need. The results of the two cases are demonstrated in Figure 5 and in Figure 6 respectively.

In the first case, there are moments when the significant change in a population's skill distribution (i.e., how many workers can reach a specific goal) will need the manager to greatly change its policy. E.g., the first two changes in Figure 5a result in new types of resources being collected; the changes in Figure 5b force the team to craft a different type of top-level item. In such cases, our agent-wise $\epsilon$-greedy exploration significantly improves the learning efficiency and increases the converged rewards. When the change is moderate, the policy learned by ours is fairly stable.

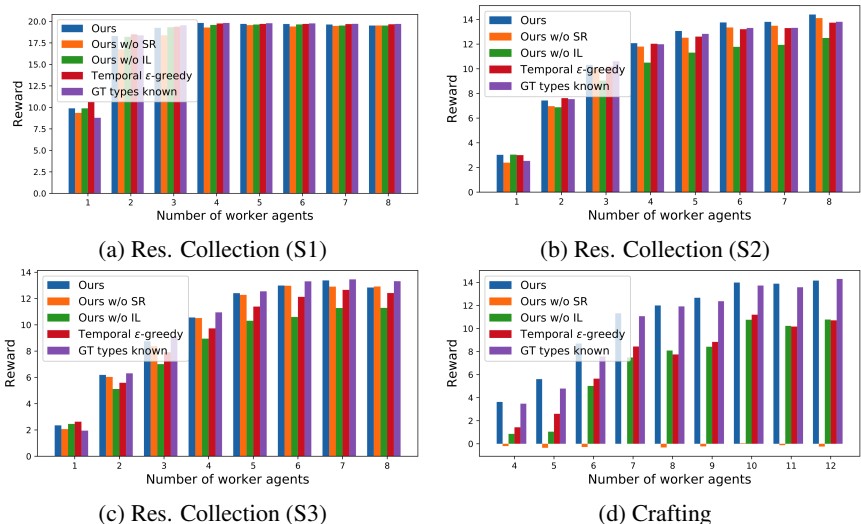

(a) Res. Collection (S1)  (b) Res. Collection (S2)

(c) Res. Collection (S3)  (d) Crafting

Figure 7: Average rewards when different numbers of worker agents are present. The policies are trained with 4 worker agents in Resource Collection and with 8 worker agents in Crafting.

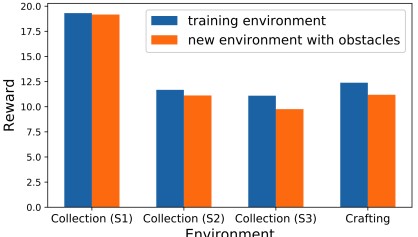

Figure 8: Testing in novel environments.  Figure 9: Performance with random actions.

In the second case, the managers trained by the three methods achieve similar converged rewards in training. While the converged reward of our approach is slightly lower than the upper bound due to exploration, it allows the manager to quickly adapt itself to a new team where it has never seen the most team members. The temporal $\epsilon$-greedy on the other hand never achieves a comparable reward even though its performance is comparable to ours when managing a fixed population.

We also want the manager's policy to have good generalization in novel scenarios unseen in training, which, in our problems, has two aspects: i) generalization in different numbers of present worker agents, and ii) generalization in new environments. It can be seen from Figure 7 that as the number of workers increases, the manager achieves higher reward until it hits a plateau. Our approach consistently performs better in all settings. It even gains higher rewards than the one with ground-truth does when there are fewer workers. We also add a few walls to create novel environments unseen in training. With the additional obstacles, workers' paths become more complex, which increases the difficulty of inferring their true minds. As suggested by Figure 8, the performance indeed decreases the most in S3 of Resource Collection where online intention inference is critical as the workers do not have fixed preferences.

So far, we have only considered rule-based worker agents with deterministic plans. To see if our approach can handle stochastic and sub-optimal worker policies, we may randomize certain amount of actions taken by the workers (Figure 9) and train a manager with these random policies. When the randomness is moderate (e.g., $\leq 20\%$), the performance is still comparable to the one without random actions. As randomness increases, we start to see larger decrease in reward. In Crafting specifically, random policies make the workers unlikely to achieve assigned goals within the time limit, thus the manager may never get top-level items if the policies are too random.

**More results**. In addition to the main experimental results discussed above, we further test our approach from different perspectives: i) showing the effect of the minimum valid period of a contract

(i.e., constraints for the manager's commitment), ii) multiple bonus levels, and iii) training RL agents as workers. We summarize these results in Appendix C.

## 6 CONCLUSIONS

In this paper, we propose Mind-aware Multi-agent Management Reinforcement Learning ($M^3RL$) for solving the collaboration problems among self-interested workers with different skills and preferences. We train a manager to simultaneously infer workers' minds and optimally assign contracts to workers for maximizing the overall productivity, for which we combine imitation learning and reinforcement learning for a joint training of agent modeling and management policy optimization. We also improve the model performance by a few techniques including learning high-level successor representation, agent-wise $\epsilon$-greedy exploration, and agent identification based on performance history. Results from extensive experiments demonstrate that our approach learns effectively, generalizes well, and has a fast and continuous adaptation.

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

## A    DETAILS OF APPROACH

### A.1    HIGH-LEVEL SUCCESSOR REPRESENTATION

We define two vectors indicating the goal achievement and bonus payment at time $t$: $\phi^g(c^t) = (\sum_i \mathbb{1}(s_i^{t+1} = s_g)\mathbb{1}(g_i^t = g) : g \in \mathcal{G})$ and $\phi^b(c^t) = (\sum_i \mathbb{1}(s_i^{t+1} = s_{g_i^t})\mathbb{1}(b_i^t = b) : b \in \mathcal{B})$. Let $\boldsymbol{w} = (b : b \in \mathcal{B})$ be the weights for different bonus payments, then the reward for the manager at the current moment can be written as $r^t = \boldsymbol{v}^\top \phi^g(c^t) - \boldsymbol{w}^\top \phi^b(c^t)$. Following the typical SR definition, we define our high-level SR as

$$\Phi^g(c^t) = \mathbb{E}\left[\sum_{\tau=0}^{\infty} \gamma^\tau \phi^g(c^{t+\tau})\right], \tag{3}$$

and

$$\Phi^b(c^t) = \mathbb{E}\left[\sum_{\tau=0}^{\infty} \gamma^\tau \phi^b(c^{t+\tau})\right]. \tag{4}$$

Thus, the value function can be written as

$$V(c^t) = \boldsymbol{v}^\top \Phi^g(c^t) - \boldsymbol{w}^\top \Phi^b(c^t). \tag{5}$$

### A.2    DETAILS OF LEARNING

The policy gradient for the goal assignment is:

$$\nabla_{\theta_g} J(\theta_g) = \frac{1}{N} \sum_{i=1}^{N} \nabla_{\theta_g} \left[\log \pi^g(g_i^t | s_i^t, m_i^{t-1}, h_i, c^t; \theta_g) A(c^t) + \lambda \mathcal{H}(\pi^g)\right], \tag{6}$$

where $A(c^t)$ is the advantage estimation defined as $A(c^t) = \sum_{\tau=0}^{\infty} \gamma^\tau r^{t+\tau} - (\boldsymbol{v}^\top \Phi^g(c^t) - \boldsymbol{w}^\top \Phi^b(c^t))$ and $\mathcal{H}(\cdot)$ is the entropy regularization weighted by the constant $\lambda = 0.01$ for encouraging exploration. Similarly, the policy gradient for the bonus assignment is

$$\nabla_{\theta_b} J(\theta_b) = \frac{1}{N} \sum_{i=1}^{N} \nabla_{\theta_b} \left[\log \pi^b(b_i^t | s_i^t, m_i^{t-1}, h_i, c^t; \theta_b) A(c^t) + \lambda \mathcal{H}(\pi^b)\right]. \tag{7}$$

The successor representations may be updated by the following gradient:

$$\nabla_{\theta_{\Phi^g}} \frac{1}{2} \left(\sum_{\tau=0}^{\infty} \gamma^\tau \phi^g(c^{t+\tau}) - \Phi^g(c^t; \theta_{\phi^g})\right)^2 + \nabla_{\theta_{\Phi^b}} \frac{1}{2} \left(\sum_{\tau=0}^{\infty} \gamma^\tau \phi^b(c^{t+\tau}) - \Phi^b(c^t; \theta_{\phi^b})\right)^2. \tag{8}$$

For imitation learning, we use the following cross-entropy loss:

$$\mathcal{L}_{\text{IL}} = \mathbb{E}\left[-\frac{1}{N} \sum_{i=1}^{N} \log \hat{\pi}(a_i^t | s_i^t, g_i^t, b_i^t, m_i^{t-1})\right]. \tag{9}$$

Note that the gradient from IL will be combined with Eq. 6, Eq. 7, and Eq. 8 to update corresponding parameters (see Algorithm 2 in Appendix B for details).

In tasks where there the unknown dependency may introduce a large cost in the beginning of training, the manager's exploration may be restricted as the policy becomes too conservative in spending, which is common in many real world scenarios. To encourage exploration in these tasks, we adopt a two-phase learning curriculum. Namely, we optionally conduct a warm-up phase before the standard learning described above. In this warm-up phase, we give loans to the manager to cover its cost (i.e., setting the total payments to be zero when optimizing the networks). In practice, we apply this only to Crafting, where we set a fixed number of episodes at the beginning of training to be the warm-up phase. Note that this only apply to the optimization; we still need to deduct the payments from the rewards as the actually outcomes (this is equivalent to paying back the loans).

---

**Algorithm 1** Rollout($T_{\max}, T_c, \epsilon, \{\mathbb{P}_i : i = 1, \cdots, N\}$)

---

**Input:** Maximum steps $T_{\max}$, commitment constraint $T_c$, exploration coefficient $\epsilon$, and the performance history of the present worker agents $\{\mathbb{P}_i : i = 1, \cdots, N\}$

**Output:** Trajectories of all workers $\{\Gamma_i^T : i = 1, \cdots, N\}$, and the rewards for the manager $R$

 1: Initialize the environment
 2: Set performance history update rate $\eta = 0.1$.
 3: $t \leftarrow 0$
 4: $\Gamma_i^1 \leftarrow \emptyset, \forall i = 1, \cdots, N$
 5: **repeat**
 6:     Observe current states $(s_i^t, a_i^t), \forall i = 1, \cdots, N$, from the environment
 7:     Encode $\mathbb{P}_i$ to $h_i, \forall i = 1, \cdots, N$
 8:     **if** $t = 0$ **then**
 9:         **for** $i = 1, \cdots, N$ **do**
10:             $\Gamma_i^0 \leftarrow \emptyset$
11:             Sample a random goal $g_i^0 \sim \mathcal{G}$, and set a minimum bonus $b_i^0$ as the initial contract.
12:             $e_i \sim U(0, 1)$
13:             Assign the contract $(g_i^0, b_i^0)$ to worker $i$ and receive $d_i^t$ (signing decision)
14:             $\tau_i \leftarrow d_i^t$
15:             $\Gamma_i^t \leftarrow \{(s_i^t, a_i^t, g_i^t, b_i^t)\}$
16:         **end for**
17:         $r^t \leftarrow 0$
18:     **else**
19:         $r^t = \sum_{g \in \mathcal{G}} \sum_{i=1}^N \mathbb{1}(s_i^t = s_g)\mathbb{1}(g = g_i^{t-1})(v_g - b_i^{t-1})$
20:         $m_i^{t-1} \leftarrow M(\Gamma_i^{t-1}, h_i), \forall i = 1, \cdots, N$
21:         $c^t \leftarrow C(\{(s_i^t, m_i^{t-1}, h_i) : i = 1, \ldots, N\})$
22:         **for** $i = 1, \cdots, N$ **do**
23:             $\tau_i' = \tau_i$
24:             **if** $(t - 1)\%T_c \neq 0$ or $e_i < \epsilon$ **then** # Commitment constraint and agent-wise $\epsilon$-greedy
25:                 $g_i^t \leftarrow g_i^{t-1}$
26:             **else**
27:                 Sample a new goal $g_i^t \sim \pi^g(\cdot|s_i^t, m_i^{t-1}, h_i, c^t)$
28:                 $\tau_i \leftarrow \tau_i \mathbb{1}(g_i^t = g_i^{t-1})$
29:             **end if**
30:             Sample a new bonus $b_i^t \sim \pi^b(\cdot|s_i^t, m_i^{t-1}, h_i, c^t)$ (w/ temporal $\epsilon$-greedy)
31:             Assign the contract $(g_i^t, b_i^t)$ to worker $i$ and receive $d_i^t$
32:             $\tau_i \leftarrow \tau_i + d_i^t$
33:             $\Gamma_i^t \leftarrow \Gamma_i^{t-1} \cup \{(s_i^t, a_i^t, g_i^t, b_i^t)\}$
34:             **if** $\tau_i' > \tau_i$ or $s_i^t = s_{g_i^{t-1}}$ **then** # the last contract was accepted and has been terminated now
35:                 $\rho_{ig^{t-1}b^{t-1}}^{\tau_i'} \leftarrow (1 - \eta)\rho_{ig^{t-1}b^{t-1}}^{\tau_i'} + \eta \mathbb{1}(s_i^t = s_{g_i^{t-1}})$
36:             **end if**
37:         **end for**
38:     **end if**
39:     $R \leftarrow R \cup \{r^t\}$
40:     $t \leftarrow t + 1$
41: **until** $t = T_{\max}$ or the task is finished
42: $T \leftarrow t$

---

**Algorithm 2** Learning Algorithm

---

 1: Initialize parameters
 2: Set the maximum steps of an episode to be $T_{\max}$, maximum training episodes to be $N_{\text{train}}$, and the number of worker agents in an episode to be $N$
 3: The coefficient for the agent-wise $\epsilon$-greedy exploration to be $\epsilon$
 4: Initialize a population of worker agents and set their performance history $\mathbb{P}$ to be all zeros.
 5: **for** $i = 1, \cdots, N_{\max}$ **do**
 6:     Sample $N$ worker agents from the training population and obtain their performance history $\{\mathbb{P}_i : i = 1, \cdots, N\}$
 7:     # Run an episode
 8:     $\{\Gamma_i^T : i = 1, \cdots, N\}, R \leftarrow$ Rollout($T_{\max}, T_c, \epsilon, \{\mathbb{P}_i : i = 1, \cdots, N\}$)
 9:     Update parameters based on the IL loss $\mathcal{L}_{IL}$ defined in Eq. (9) and the gradients defined Eq. (6), Eq. (7), and Eq. (8) jointly.
10: **end for**

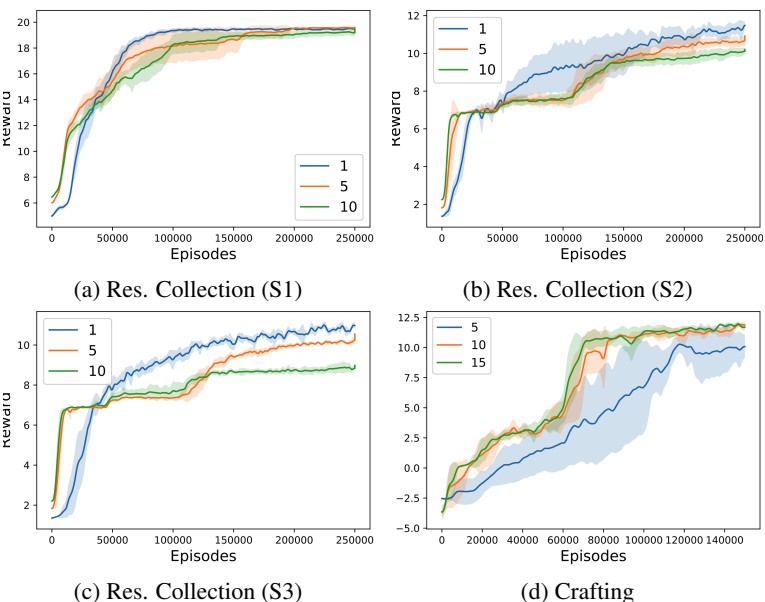

(a) Res. Collection (S1)  (b) Res. Collection (S2)

(c) Res. Collection (S3)  (d) Crafting

Figure 10: Learning curves in three settings of Resource Collection and in Crafting when applying different commitment constraints for the manager. The numbers indicate how many steps a contract must holds.

## B  PSEUDO CODE OF OUR ALGORITHMS

We summarize the rollout algorithm and the learning algorithm in Algorithm 1 and Algorithm 2 respectively.

## C  MORE EXPERIMENTAL RESULTS

### C.1  CONSTRAINING THE MANAGER'S COMMITMENT

The manager's commitment is defined as the shortest time a contract must remain unchanged, which essentially constrains how frequent the manager can change its goal and bonus assignment. While short-term commitment allows the manager to quickly update contracts once it has a better mind estimation or once a goal has been reached, long-term commitment often leads to a more accurate skill assessment when the tasks are difficult (e.g., crafting high-level items depends on the results of other tasks and thus needs a longer time). This is supported by the results in Figure 10: shorter commitment works better in Resource Collection while Crafting needs a longer commitment. Note that the commitment constraint is 1 step and 10 steps for Resource Collection and Crafting respectively in all other experiments.

### C.2  MULTIPLE BONUS LEVELS

In previous experiments, the internal utility of goals for a worker agent is either 0 or 1. Here, we sample the internal utility from 0 to 3. Consequently, the manager needs to select the right bonus from multiple choices to pay each worker (i.e., a bonus from 1 to 4 for Resource Collection and a bonus from 0 to 4 for Crafting). In Resource Collection, the manager will get a reward of 5 for every collected resource; in crafting, the reward for a top-level item is still 10. As shown in Figure 11, the advantage of our approach is even more significant compared to the ones in single payment level.

### C.3  RL AGENTS AS WORKERS

Finally, we train a population of 40 RL worker agents for Resource Collection, where each one is trained with only one goal, and for each goal we train 10 agents using different random seeds.

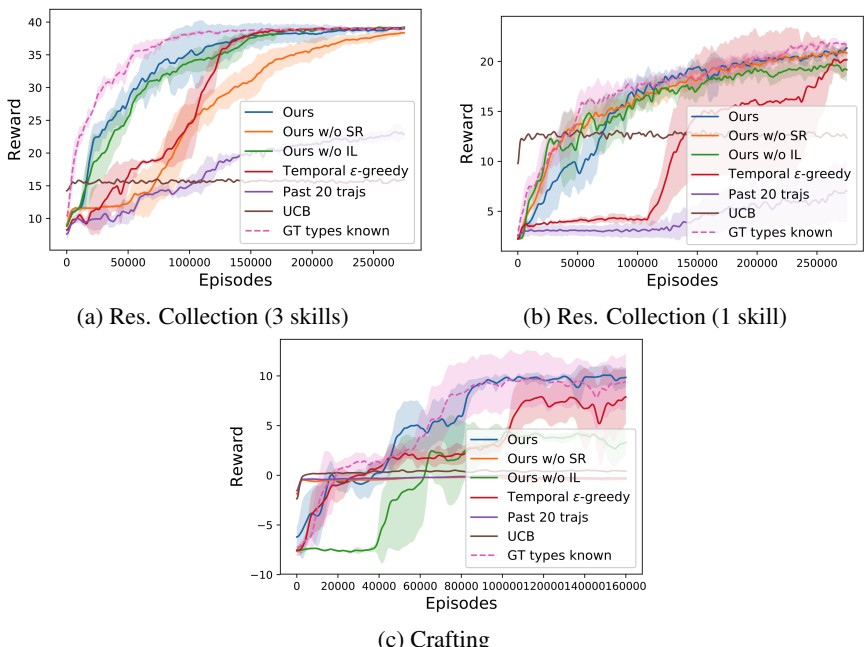

(a) Res. Collection (3 skills)  (b) Res. Collection (1 skill)

(c) Crafting

Figure 11: Learning curves when there are multiple bonus levels. In (a), a worker can reach 3 goals; in (b), a worker can reach 1 goal; in (c), a worker can collect raw materials and craft one type of items.

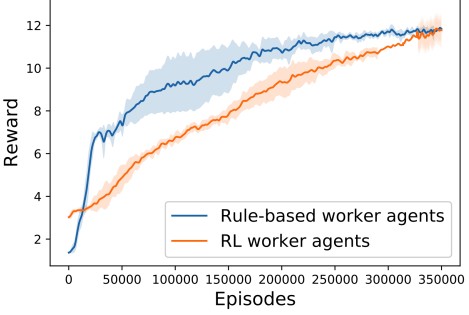

Figure 12: Comparing training with rule-based worker agents and with RL worker agents in Resource Collection.

This creates a population with similar skill distributions as in S2, but with very different policies. Figure 12 suggests that training to manage RL agents is slower as their policies are less predictable and less rational, but our approach can still gradually learn a good policy whose performance is comparable to the one using rule-based worker agents.

# D IMPLEMENTATION DETAILS

## D.1 NETWORK ARCHITECTURE

**Performance History Module**. We flatten the matrices in worker's performance history $\mathbb{P}_i$ and concatenate them together to get a single vector. We then encode this vector into a 128-dim history representation $h_i$.

**Mind Tracker Module**. We represent the state of a worker by multiple channels corresponding to different types of items. We also use four channels to indicate its orientation. We augment the state with additional $|\mathcal{A}||\mathcal{G}||\mathcal{B}|$ channels, where a channel is either all ones or all zeros for indicating the action it takes, and the goal and bonus it receives. We then encode the state into a 128-dim hidden state by a convolutional layer with 64 channels and kernels of $1 \times 1$, a fully connected (FC) layer (128-dim), and an LSTM with 128 hidden units. We fuse this vector with the history representation $h_i$. Specifically, we adopt an attention-based mechanism for the fusion, where we first get an attention vector (128-dim) from the history representation by an FC layer with sigmoid activation, and then do element-wise product between the attention vector and the hidden state from the LSTM. The fused vector becomes $m_i^t$. This can be formally written as $m_i^t = f(\ell_i^t, h_i) = \ell_i^t \odot \sigma(h_i)$, where $\sigma(\cdot)$ is an FC layer with sigmoid activation, $\ell_i^t$ is the hidden state from the LSTM, and $\odot$ is element-wise product. We fuse it with the state using the same mechanism: $f(\phi(s_i^{t+1}, g_i^{t+1}, b_i^{t+1}), m_i^t) = \phi(s_i^{t+1}, g_i^{t+1}, b_i^{t+1}) \odot \sigma(m_i^t)$, where $\phi(s_i^{t+1}, g_i^{t+1}, b_i^{t+1})$ is the state encoding. By feeding the fused vector to an FC layer with softmax activation, we may get the predicted worker policy.

**Manager Module**. For each worker, we concatenate its mind representation and history representation together and fuse it with the worker's state using the attention-based mechanism where the attention vector comes from the concatenated vector. By pooling over these fused vectors of individual workers, we can get the context vector, from which we construct the two successor representations by two separate FC layers. Here, we use average pooling, but one may also use other pooling mechanisms. Finally, for each worker, we concatenate the context vector with its fused vector we obtained before pooling, and consequently get the goal policy and bonus policy by two FC layers with softmax activation.

All modules are trained with RMSProp (Tieleman & Hinto, 2012) using a learning rate of 0.0004.

## D.2 RULE-BASED WORKER AGENTS

Each rule-based worker finds a shortest path to the nearest location related to a goal, and its skill is defined as the post effect of its "collect" and "craft" actions. In particular, for collecting certain resource/material, it will go to the closest one that has the same type as the goal indicates and is currently not being collected by other agents, whereas for crafting an item, it will go to the corresponding work station if it is currently unoccupied. If a worker can perform collecting tasks, then after it takes "collect" action, the item will be collected from the map and appears in the inventory; otherwise no real effect will appear. This applies to crafting tasks as well, except in crafting, task dependencies must also be satisfied before "craft" action can take real effect.

When considering random actions, for each step, we sample a random action with the specified chance to replace the action from the rule-based plan.

## D.3 RL WORKER AGENTS

We implement all RL worker agents using the same network architecture, where an agent's state is augmented by additional channels to include the reward for each goal (i.e., $|\mathcal{G}||\mathcal{B}|$ channels). We use a convolution layer with 64 channels and kernels of $1 \times 1$ to encode the state, and feed it to an 128-dim FC layer and then an LSTM with a 128-dim hidden state. We then predict the policy using

an FC layer with softmax activation based on the hidden state from the LSTM. For each goal, we train 10 RL worker agents using 10 random seeds. For each episode, we randomly assign a reward from $b \in \mathcal{B}$ to an agent as the hypothetical reward it may receive from a manager. We then set the corresponding channel to be all ones and set the remaining $|\mathcal{G}||\mathcal{B}| - 1$ channels to be all zeros. Note that we assume all RL workers have the ability to perform "collect" and "craft" actions.

