# OpenReview forum: "M^3RL: Mind-aware Multi-agent Management Reinforcement Learning"
_ICLR.cc/2019/Conference_

### Official Review · AnonReviewer1 · 2018-10-29
**Multi-agent Management using RL**

**Rating:** 6
**Confidence:** 1

**Review:**

This paper studies the problem of coordinating many strategic agents with private valuation to perform a series of common goals. The algorithm designer is a manager who can assign goals to various agents but cannot see their valuation or control them explicitly. The manager has a utility function for various goals and wants to maximize the total revenue. The abstract problem is well-motivated and significant and is an entire branch of study called algorithmic mechanism design. However often many assumptions have to be made to make the problem mathematically tractable. In this paper, the authors take an empirical approach by designing an RL framework that efficiently maximizes rewards across many episodes. Overall I find the problem interesting, well-motivated. The paper is well-written and contains significant experiments to support its point. However, I do not have the necessary background in the related literature to assess the significance of the methods proposed compared to prior work and thus would refrain from making a judgment on the novelty of this paper in terms of methodology. Here are some of my comments/questions to the author on this paper.


(1) I want to clarify how the skills of the agents play a role in the problem setup. Does it show up in the expression for the manager's reward? In particular, does it affect the Indicator for whether a goal is completed Eq. (2) via a process that need not be explicitly modeled but can be observed via a feedback of whether or not the goal is completed? So in the case of resource collection example, the skill set is a binary value for each resource, whether it can be collected or not?

(2) Related to the first point, the motivation for modeling the agents as maximizing their utility is the assumption that agents do not know their skills. I am wondering, is this really justified? Over the course of episodes, can the agents learn their skills based on the relationship between their intention and the goals they achieve? In the resource collection example, when they reach a resource and are not able to collect it, they understand that they do not have the corresponding skill. Is there a way to extrapolate the results from this paper to such a setting?

(3) I am slightly concerned about the sample complexity of keeping track of the probability of worker i finishing goal g within t steps with a bonus b. This scales linearly in parameters which usually would be large (such as the number of time-steps). Are there alternate ways to overcome maintaining the UCB explicitly, especially for the number of time-steps?

Some minor comments on the presentation.

(1) What are the units for rewards in the plots? Is it the average per episode reward? It would be good to mention this in the caption.

(2) There are a few typos in the paper. Some I could catch was,

- Last line in Page 5: "quantitative" -> "quantity"
- Page 8: skills nad preferences -> skills and preferences
- Page 8: For which we combining -> for which we combine

---

> ### Author Response · Authors · 2018-11-21
> **Author response**
>
> Thank you for your reviews.  Here are our responses to your questions:
>
> 1. Clarify how the skills of agents play a role in the problem setup
> We clarify the definition of skills and how it influences the manager’s decision as follows.
>
> i) As defined in Section 3, an agent’s skill depends on its state transition probabilities and its policy. The state transition probabilities define if a resource can be collected by an agent (i.e., whether the “collect” action executed by this agent will have real effect), and it is equivalent to a binary value for each resource in Resource Collection. The agent’s skill also depends on its policy because it affects how fast an agent can achieve a goal. E.g., when the agent has a suboptimal policy, it may not be able to reach a goal within the time limit even though it actually can collect the resource if given more time.
>
> ii) The skills are completely hidden from the manager. It can be inferred by the manager based on the performance history, and also on the estimated worker policies by IL. However, only checking whether a goal is reached is not sufficient to determine skills. Failing to reach a goal may be a result of several reasons -- it may be because i) the bonus in the contract is too low, ii) the contract terminates prematurely before the agent can reach the goal, or iii) the assigned task depends on another task which has not been finished yet. So the manager needs to infer agents’ skills, preferences, and the task dependency jointly through multiple trials.
>
> 2. Is maximizing utility justified?
> Maximizing utility is actually the setup in similar problems in economics. Just like those problems (e.g., mechanism design), this paper focuses on scenarios where agents won’t truthfully or clearly reveal its skills and preferences to the manager, and do not always behave optimally. As we stated in the paper, maximizing utility is more realistic, and typically the span of the decision making process of the manager is much shorter than the time needed for improving worker agents. Let’s consider a simple scenario. An agent is unable to collect a certain kind of resource. By maximizing its utility, it may still accept the contract and go to that resource. Once a resource is occupied by this agent, other agents can no longer collect it according to our setting. This means that the resource will never be really collected.
>
> As an empirical evidence,  you may compare the S2 and S3 settings with S1 in Resource Collection. In S2 and S3, workers may prefer a task that it can not perform, which should never happen in the case of maximizing return. As a result (shown in Figure 4b and Figure 4c), the training difficult significantly increases.
>
> 3. Are there alternate ways to overcome maintaining the UCB explicitly, especially for the number of time-steps?
> Yes, there are ways to overcome this. First, we can define small time intervals instead of maintaining statistics for each step (i.e., combining statistics in every dT consecutive steps will reduce the complexity to 1 / dT of the original size). Note that this has been done in results shown in Appendix C.1, where dT also means that for every dT steps, the manager can only change the contracts once. Second, we may define a maximum number of steps to be considered in the performance history, which can be determined by the upper bound of the execution time for a subtask, and can be smaller than the step limit of the whole episode.
>
> 4. What are the units for rewards in the plots?
> It is the average per episode. The reward is defined as in Section 5.1.1 and Section 5.1.2 without any rescaling. We have added this in the caption.
>
> 5. Typos
> Thank you for pointing out these typos. We will fix them in the next revision.

---

### Official Review · AnonReviewer2 · 2018-11-03
**An interesting exploration of RL for principle-agent problems**

**Rating:** 6
**Confidence:** 3

**Review:**


This paper studies the problem of generating contracts by a principal to incentive agents to optimally accomplish multiagent tasks. The setup of the environment is that the agents have certain skills and preferences for activities, which the principal must learn to act optimally. The paper takes a combined approach of agent modeling to infer agent skills and preferences, and a deep reinforcement learning approach to generate contracts. The evaluation of the approach is fairly thorough.

The main novel contribution of the paper is to introduce the principal-agent problem to the deep multiagent reinforcement learning literature.

My concerns are:
- The paper should perform a literature search on related work from operations research, including especially principal-agent problems, which are not currently surveyed, and perhaps also optimal scheduling problems.
- How do the problems introduced either map onto real applications or map onto environments studied in existing literature (such as in operations research)?
- More details should be given on the mind tracker module.
- Is it necessary to use deep reinforcement learning for contract generation?  If the agent modeling is good, the optimal contracts look like they are probably simple to compute directly in the environments studied.

Overall, the paper is somewhat interesting and relatively technically sound, but the contribution seems marginal. The problems studied seem pulled out a hat, when they could be situated in specific existing literature.

---

> ### Author Response · Authors · 2018-11-21
> **Author response**
>
> Thank you for your comments and suggestions. We respond to your questions and concerns as follows.
>
> 1. Connection with principal-agent problems.
> Thank you for pointing this out. We really appreciate it. The problem we address is indeed closely connected to principal-agent problems, or moral hazard problems in economics, which considers whether the agent makes the best choice for what the principal delegates (e.g., a plumber might make more money by suggesting an overhaul rather than a short-term fix). In this setting, there are a lot of issues to be modeled, e.g., information asymmetry between principals and agents, how to setup incentive cost, how to infer agents’ types and how to monitor their behaviors, etc. Traditional approaches [1] in economics build mathematical models to address these issues separately, leading to complicated models with many tunable parameters. In comparison, our paper provides a practical end-to-end computational framework to address this problem in a data-driven way, once the agents’ utility function is written down as a combination of principal’s request and its own preference (Eqn. 1). Moreover, this framework is adaptive to changes of agents’ preferences and capabilities, which very few papers in economics have addressed.
>
> Because of the connection to principal-agent problems and the data-driven nature of the proposed method, there could be a broad number of practical applications.
>
> We will incorporate a more thorough literature reviews in the next revision.
>
> [1] The theory of incentives: the principal-agent model, Jean-Jacques Laffont, 2001
>
> 2. More details should be given on the mind tracker module.
> We will explain more implementation details in the appendix in the next revision. We will also release the code.
>
> 3. Is it necessary to use deep reinforcement learning for contract generation?
> As stated in the introduction, one of the main points of this work is about incomplete information. I.e., we do not know the true agent models and their mental states, and also do not assume that the task dependency is known. In real world problems, we indeed can not assume that a manager knows the exact nature of other agents. So we want to train a manager that can quickly model worker agents through observations and simultaneously generate optimal contracts. In contrast, traditional methods do not consider task dependency, and usually assume agent types are either known or follow a given distribution. Also, deep models are flexible enough to handle complicated interactions between agents and changes of settings. Thus, deep RL is a more suitable approach than traditional methods under the incomplete information setting.

---

### Official Review · AnonReviewer3 · 2018-11-08
**Multi agent RL with self interesting agents: New formulation and solution in simple environments.**

**Rating:** 7
**Confidence:** 4

**Review:**

Summary:

This paper proposes a way to train a manager agent which would manage a bunch of worker agents to achieve a high-level goal. Each worker has its own set of skills and preferences and the manager tries to assign sub-tasks to these agents along with bonuses such that the agents can even perform tasks that are not preferred by them. Authors achieve this by training a manager which tracks the skills and preferences of the agents on the fly. Authors have done an extensive analysis of the proposed approach in two simple domains: resource collection and crafting.

Major comments:

This paper focuses on multi-agent settings with self-interested agents. The problem formulation and the solution are novel enough. Experiments are on toy domains with very few goals and sub-task dependencies. However, authors have done a good job in doing an extensive analysis of the proposed approach.

1.	Can you comment about the scalability of the proposed solution when the number of possible subtasks increases? When the sub-task dependency graph size increases?

2.	What is the reason for using rule-based agents in all the experiments? It would have been more useful if all the analysis are done with RL agents rather than rule-based agents. It would also make the paper stronger.

3.	Are the authors willing to release the code? Overall the model looks complicated and the appendix is not sufficient to reproduce the results in the paper. I would increase my rating if the authors are willing to release the code to reproduce all the results reported in the paper.


Minor comments:

1.	Page 3, line 9: “typical” -> “typically”
2.	Page 3, “intention” section: “Based on the its reward ..” Check grammar.
3.	Page 5, last line: “the total quantitative is 10” check grammar.
4.	Page 8, conclusions, second line: “nad” -> “and”
5.	Page 8, conclusions, 4th line: “combing” -> “combine”

---

> ### Author Response · Authors · 2018-11-21
> **Author response**
>
> Thank you for your reviews and comments. We address your questions as follows.
>
> 1. Scalability of the proposed solution
> From our current results, you may see that our approach has a decent scalability -- even though we doubled the subtasks and also introduced additional dependency in Crafting compared to Resource Collection, it does not need much more episodes for converging to optimal policies, where our agent-wise exploration plays an important role. Generally speaking, deploying more present workers coupled with our agent-wise exploration should significantly improve the learning efficiency and overcome the challenges introduced from more substasks or a larger dependency graph. In addition, the computational complexity is linear in terms of the number of agents, so our approach is also scalable when there are more agents.
>
> 2. What is the reason for using rule-based agents in all the experiments?
> We have actually used RL agents as well (Appendix C.3), and it showed that our approach also works when workers are RL agents. In the main results, we focus on rule-based agents because it is computationally demanding to train a large population of RL agents, and our focus was not about the worker policies but rather how the manager assesses the workers’ mental states and encourages an optimal collaboration accordingly. In this paper, using a cheap rule-based implementation with randomness has demonstrated the effect of different components of our approach.
>
> 3. Are the authors willing to release the code?
> Yes, we do plan to open source our implementation. Specifically, the game environment and the worker agents were implemented in Python and it runs at a speed of more than 300 steps per second. We used PyTorch as the framework for implementing all the network modules. Typically it took < 10 hours to get a converged result by our approach on a single Nvidia Tesla V100 GPU.
>
> 4. Typos
> Thanks for pointing out these typos. We will fix them in the next revision.

---

> > ### Comment · AnonReviewer3 · 2018-11-26
> > **Post-rebuttal comments**
> >
> > I am ok with the rebuttal. Even though it would have been interesting to have neural net based agents, I understand authors' computational constraints.
> >
> > I am increasing my rating from 6 to 7.

---

### Author Response · Authors · 2018-11-26
**Submission revision**

We thank all reviewers for their comments and suggestions.  We have revised our submission and also plan to release our code. In the revision, we i) discussed the connection between our work and principal-agent problems including how our work differs from the traditional approaches in classical principal-agent problems, ii) added implementation details about the mind tracker module, and iii) fixed the typos.

---

> ### Comment · AnonReviewer1 · 2018-11-26
> **Thanks!**
>
> Thanks a lot for your rebuttal. We have gone over your rebuttal/revised submission.

---

### Meta-Review · Area_Chair1 · 2018-12-14
**Deep reinforcement learning for principle agent problems**

**Confidence:** 4
**Recommendation:** Accept (Poster)

**Metareview:**

The paper addresses a variant of multi-agent reinforcement learning that aligns well with real-world applications - it considers the case where agents may have individual, diverging preferences. The proposed approach trains a "manager" agent which coordinates the self-interested worker agents by assigning them appropriate tasks and rewarding successful task completion (through contract generation). The approach is empirically validated on two grid-world domains: resource collection and crafting. The reviewers point out that this formulation is closely related to the principle-agent problem known in the economics literature, and see a key contribution of the paper in bringing this type of problem into the deep RL space.

The reviewers noted several potential weaknesses: They asked to clarify the relation to prior work, especially on the principle-agents work done in other areas, as well as connections to real world applications. In this context, they also noted that the significance of the contribution was unclear. Several modeling choices were questioned, including the choice of using rule-based agents for the empirical results presented in the main paper, and the need for using deep learning for contract generation. They asked the authors to provide additional details regarding scalability and sample complexity of the approach.

The authors carefully addressed the reviewer concerns, and the reviewers have indicated that they are satisfied with the response and updates to the paper. The consensus is to accept the paper.

The AC is particularly pleased to see that the authors plan to open source their code so that experiments can be replicated, and encourages them to do so in a timely manner. The AC also notes that the figures in the paper are very small, and often not readable in print - please increase figure and font sizes in the camera ready version to ensure the paper is legible when printed.

---

> ### Author Response · Authors · 2019-03-06
> **Code released.**
>
> As promised, we have released the code of this paper:
>
> https://github.com/facebookresearch/M3RL
>
> Thanks all for the interest!